# Real-World Impact of Pneumococcal Conjugate Vaccines on Vaccine Serotypes and Potential Cross-Reacting Non-Vaccine Serotypes

**DOI:** 10.3390/vaccines13060651

**Published:** 2025-06-17

**Authors:** Kevin Apodaca, Lindsay R. Grant, Johnna Perdrizet, Derek Daigle, Gabriel Mircus, Bradford D. Gessner

**Affiliations:** 1Respiratory Vaccines and Antivirals, Pfizer Chief Medical Affairs Office, New York, NY 10965, USA; 2HTA, Value and Evidence, Global Access and Value, New York, NY 10001, USA; 3US Medical Affairs, Vaccines and Antivirals, New York, NY 10001, USA

**Keywords:** *Streptococcus pneumoniae*, pneumococcal conjugate vaccines, invasive pneumococcal disease, serotype epidemiology

## Abstract

Background: Clinical trials and serological studies have demonstrated that vaccine-induced antibodies can cross-react with some non-vaccine serotypes. However, there are limited longitudinal data on the impact of pneumococcal conjugate vaccines (PCV) on cross-reactive serotypes after implementation in immunization programs. This study examines the impact of PCVs on pneumococcal disease cases due to potential cross-reactive serotypes. Methods: Eleven countries with serotyped invasive pneumococcal disease (IPD) surveillance data that had introduced PCV10 or PCV13 were identified. The analysis focused on IPD cases due to serotypes included in PCV10 and PCV13 (PCV10/13 VTs: 6B, 9V, 19F, 23F; PCV13 only VTs: 6A, 19A) and to vaccine-related serotypes (VRTs: 6C, 9N, 23A, 23B) that may be immunologically related to VTs in children under 5 years old. For each country, the number of IPD cases were charted over time according to serogroup. Results: Following PCV introduction, reductions in VT IPD cases were observed in all countries, while some VRT IPD cases remained unchanged or increased. Serotype 19A cases declined in PCV13 countries but increased in countries that introduced PCV10. VRT 6C cases rose in PCV10 countries but showed minimal change in PCV13 countries. In PCV13 countries, 9N cases remained unchanged while 23A and 23B experienced modest increases. Conclusions: The inclusion of VT 19A in PCV13, but not in PCV10, may account for the significant increase in VRT 19A cases in PCV10 countries. The slight change in VRT 6C cases in PCV13 countries compared to the significant rise in PCV10 countries suggests that PCV13 provides cross-protection for serotype 6C through serotype 6A. Cross-protection could not be determined for other VRTs, as their cases increased or remained unchanged or had insufficient data for evaluation.

## 1. Introduction

Starting in the early 2000s, pneumococcal conjugate vaccines (PCVs) were implemented in pediatric national immunization programs (NIPs) to provide protection against pneumococcal disease, which is a leading cause of morbidity and mortality among children <5 years of age [1]. A heptavalent PCV (PCV7) was the first PCV introduced, covering the serotypes responsible for the majority of invasive disease in children at the time, including serotypes 4, 6B, 9V, 14, 18C, 19F, and 23F. A few years after the introduction of PCV7, higher-valent PCVs were later developed, namely the 10-valent vaccine (PCV10) and the 13-valent vaccine (PCV13). Both vaccines covered additional serotypes (1, 5, and 7F in PCV10, and 1, 3, 5, 6A, 7F, and 19A in PCV13) as well as original seven serotypes included in PCV7.

Pneumococcal serotypes are defined by their unique chemical composition and immunological properties of the capsular polysaccharide of *Streptococcus pneumoniae*, and these serotypes can be categorized into serogroups based on similar antigenic properties, particularly the structures of their surface antigens. Due to such similarities in structure, antigens of certain vaccine serotypes may elicit cross-reactive antibodies to related non-vaccine serotypes, which are often within the same serogroup. Cross-reactivity occurs when antibodies generated against vaccine serotype antigens bind non-vaccine serotype antigens (often those within the same serogroup), potentially conferring cross-protection, resulting in prevention of disease. The concept of cross-reactivity has impacted the design, development, and evaluation of vaccines [2]. While cross-reactivity from PCVs has been reported in clinical trials and post-licensure studies [3,4,5,6,7,8,9,10,11,12,13], there are limited longitudinal data depicting potential cross-reactive serotype trends after implementation of PCVs in infant immunization programs. Cross-reactivity can potentially extend the protective effects of a vaccine beyond the included serotypes. However, the extent and effectiveness of cross-protection has not been studied in detail and will vary depending on the serotype composition of the PCV and the population’s epidemiological context. Understanding cross-reactivity is crucial for assessing the overall effectiveness of PCVs and guiding the development of future vaccines that aim to offer broader protection against pneumococcal diseases.

The goal of the study was to assess if existing data suggest that PCVs can reduce IPD due to cross reactive serotypes. This was performed by evaluating trends in pediatric IPD due to potential cross-reactive, vaccine related serotypes for countries that had implemented either PCV10 or PCV13 into their pediatric NIP. Because crude trend analyses cannot determine causation, this study should not be considered hypothesis-generating.

## 2. Methods

### 2.1. IPD Data

Countries that met the following criteria were included: (1) introduced PCV10 or PCV13 into their pediatric NIP, (2) reported annual serotype-specific IPD longitudinal case counts for children < 5 years of age, (3) had data from at least 1 year prior and at least 5 years after PCV introduction. Countries that had fewer than 20 IPD cases per year or had periodically switched between PCV10 and PCV13 in their NIP were excluded. Among countries that had introduced PCV10, Colombia [14,15], Finland, Brazil [14,15,16], and Paraguay [14,15] were included. Among countries that had introduced PCV13, Argentina [14,15,17], Chile [14,15], Mexico [14,18], Germany [19], Israel [20], United States [21], and South Africa [22] were included.

IPD case count data were extracted by serotype and age group from the identified data sources (Table 1). For each country, data for serotypes included in PCV7, PCV10, or PCV13 and for their potentially cross-reactive serotypes were obtained for children < 5 years of age. While countries differed in the years during which serotype specific data were available, all had data available from 1998 through 2019 that satisfied the inclusion/exclusion criteria, and consequently this was used as the study period. Data from 2020 onward were excluded because of the effect of the COVID pandemic on IPD cases.

### 2.2. Data Analysis

To improve the robustness of the results, analyses were restricted to serotypes belonging to a serogroup with more than 20 cumulative IPD cases over the study period in any PCV setting. Serotypes were grouped based on inclusion in PCV7, PCV10, or PCV13 and were defined as vaccine-type (VT) serotypes and vaccine-related (VRT) serotypes (Table 2). Countries were grouped based on whether they had introduced PCV10 or PCV13 into their respective NIP. For each country, the number of IPD cases was graphed over time by serogroup (i.e., serogroup 6: 6A, 6B, 6C; serogroup 9: 9N, 9V; serogroup 19: 19A, 19F; and serogroup 23: 23A, 23B, 23F).

Availability of serotype data varied by country. Serogroups 7 and 18 were excluded due to VRTs 7A, 7B, 18C, 18A, 18B, and 18F having little to no cases in any PCV setting during the study period. Serogroup 19 (19A, 19F) and serogroup 6 (6A, 6B, 6C) were available for included countries. Among included PCV10 countries, only Finland had data for serogroups 23 (23A, 23B, 23F) and 9 (9V, 9N). Among included PCV13 countries, data for serogroups 23 and 9 were available for most countries except Argentina and Mexico.

Because of differences in previous vaccine history among the countries included, relative changes in cases were not calculated. The implementation of PCV7 differed across countries that adopted PCV10 or PCV13, and among those that introduced PCV10 or PCV13, the duration of PCV7 use also varied.

## 3. Results

### 3.1. PCV10 Countries

In countries that introduced PCV10, IPD cases due to VTs decreased over time following PCV10 introduction, while IPD cases due to VRTs exhibited more variability in their trends (Figure 1). For instance, VT 19F cases decreased (from 4–14 1 year prior to PCV10 intro to 0–1 cases in 2018/2019), while VRT 19A cases increased (from 0–12 to 13–78 cases in 2018/2019) in PCV10 countries assessed. Among serotypes in serogroup 6 in PCV10 countries assessed, VT 6B experienced a decline in cases (from 4–34 to 0 cases in 2018/2019), while VRT 6C cases increased (from 0–3 to 1–12 cases in 2018/2019) and VRT 6A decreased (from 3–22 to 0–9 cases in 2018/2019). Decreases in VT 19F and 6B cases and increases in VRT 19A, 6A, and 6C cases were observed in Colombia, where PCV7 was introduced 3 years prior to PCV10. In Finland, VTs 23F and 9V decreased (from 6 and 2 cases to 0 and 0 cases, respectively), while lack of disease due to VRTs 23A, 23B, and 9N were maintained (0 cases in 2018/2019). No data was available for VRTs 9N, 23A, and 23B for the other PCV10 countries.

### 3.2. PCV13 Countries

In countries that introduced PCV13, IPD cases due to VTs decreased over time following PCV13 introduction, while IPD cases due to VRTs exhibited more variability in their trends (Figure 2). Cases decreased due to VTs for serotypes 19F (3–62 cases 1 yr prior to PCV13 intro to 0–14 cases in 2018/2019) and 19A (22–190 cases to 0–23 cases in 2018/2019) in the PCV13 countries assessed. Among the PCV13 countries included, IPD due to VTs 6B and 6A declined (from 0–76 cases and 0–77 cases to 0–2 cases and 0–3 cases, respectively). Serotype 6C declined in Mexico and Argentina (from 4 cases to 0 cases in 2018/2019), while there are too few cases to evaluate trends for other countries. Similar trends in cases were found serotypes among serogroup 9, where VT 9V cases decreased (0–19 cases to 0–2 cases), while VRT 9N cases had minimal change (2–4 cases in 2018/2019, respectively). VRT 23F cases decreased (1–58 cases to 0–3 cases) in all PCV13 countries assessed. In Germany and USA, a modest increase was seen for VRT 23A cases (from 0–1 to 2–6 cases in 2018/2019) and VRT 23B cases (from 2–4 cases to 9–10 cases in 2018/2019). South Africa also saw a moderate increase in VRT 23B cases (from 2 cases to 5 cases in 2019). All PCV13 countries included in this analysis had introduced PCV7, except for Argentina, which only introduced PCV13. In Argentina, cases due to VTs 19A, 19F, 6A, and 6B as well as VRT 6C declined after PCV13 introduction.

## 4. Discussion

We assessed the impact of PCVs on IPD caused by vaccine-type and vaccine-related serotypes for children < 5 years using national surveillance data from countries that introduced PCV10 or PCV13 in their pediatric NIPs. Overall, we observed a reduction in IPD cases due to VTs 6B, 9V, 23F, and 19F in countries that introduced PCV10 or PCV13 and 6A and 19A in countries that introduced PCV13. On the other hand, trends in IPD cases due to VRTs 6A (VRT for PCV10), 6C, 9N, 23A, 23B, and 19A (VRT for PCV10) varied following the introduction of PCV10 or PCV13.

Cases due to serotype 19A increased when it was not included in the vaccine formulation regardless of the presence of serotype 19F. Conversely, the reduction in 19F IPD or maintenance of low cases across countries following higher valent PCV introduction are likely due to the inclusion of 19F in PCV7, PCV10, and PCV13. The decline in 19A cases after the introduction of PCV13 (which contains this serotype and thus provides direct protection) has been extensively reported [22,23,24,25,26]. The increase in 19A cases observed in PCV10 countries, but not in PCV13 countries, suggests a lack of cross-protection by PCV10. Other studies have reported that the incidence of serotype 19A did not significantly differ between the pre- and post-PCV10 periods [27,28,29] and, in some countries, an increase in 19A cases after PCV10 introduction was observed [30,31,32,33,34]. In Belgium, a rapid reemergence of 19A IPD was reported after switching from PCV13 to PCV10 in children [35,36,37]. Such an increase in 19A disease is worrisome because it is associated with high invasive disease potential and antimicrobial resistance [38,39].

Divergent trends were also observed among serotypes 6A, 6B, and 6C. Cases due to serotypes 6A and 6B declined, while serotype 6C experienced a reduction when 6A was included in the vaccine, likely due to cross-protection. However, the moderate increase in 6C cases when only 6B was included in the vaccine indicates no cross-protection for 6C from 6B. The decline of 6B cases has been widely reported, due to direct protection provided by PCV7, PCV10, and PCV13 [40,41,42,43]. Direct protection provided by PCV13 as well as cross-protection by PCV7 has led to the decline of serotype 6A cases [42,43]. Cases due to serotype 6A have also decreased in PCV10 countries, which like PCV7 contains serotype 6B and thus likely provides some cross-protection against 6A [44]. 6C cases have also been reported to increase after PCV10 introduction in other publications [44,45,46].

Serotype 9V cases generally decreased or continued to be low after PCV introduction in both PCV10 and PCV13 settings, while 9N cases experienced minimal change. The decline of 9V cases is attributed primarily to the direct protection offered by PCV7, PCV10, and PCV13 against 9V disease in children [40,41,42]. Although serotype 9N cases experienced minimal change despite not being targeted by these PCVs, 9N cases were already low prior to PCV introduction in the countries analyzed. Other studies have reported increasing serotype 9N incidence due to serotype replacement [47].

Serotype 23F cases have generally declined or maintained low levels in both PCV10 and PCV13 countries, likely due to its inclusion in PCV7, PCV10, and PCV13. Serotype 23A cases showed minimal change in PCV10 and moderate increases in PCV13 countries. Moderate in serotype 23B cases was observed in PCV13 countries, despite evidence of moderate vaccine effectiveness of PCV13 against serotype 23B IPD among children [48]. Cross-protection may not be sufficient to suppress 23B disease. Even if a vaccine provides some protection against vaccine-related serotypes, increased exposure due to serotype replacement may mask the effect. Recent studies have reported serotypes 23A and 23B as major contributors to IPD and otitis media, highlighting the importance of monitoring these serotypes [48,49].

Immune responses of PCV7, PCV10, and PCV13 against 19A have varied. Sera of PCV7-vaccinated children has been shown to have no opsonophagocytic activity (OPA) responses against serotype 19A [50]. PCV10 was reported to elicit higher immune response against 19A compared to children vaccinated with PCV7, but the percentage of children with protective OPA levels after PCV10 vaccination did not exceed 40% after the primary series [5,51,52,53,54]. On the other hand, PCV13 has been shown to elicit a much higher OPA response against serotype 19A when compared to PCV7 and PCV10 [51,54]. The lower OPA responses induced by PCV7 and PCV10 vaccination could suggest a lack of or limited cross-protection of these vaccines against 19A disease, which is consistent with increasing cases of 19A after PCV7 and PCV10 introduction. PCV13 vaccination has also been found to induce cross-reactive functional antibody against 6C, which is likely due to the structural and immunologic similarities between serotypes 6A and 6C [46]. Studies in the US among children have shown high levels of 6C OPA after PCV13 vaccination, but not after PCV7 [46,55,56]. Given the lower 6C OPA and the increasing cases of 6C after PCV7 and PCV10 introduction, cross-protection against 6C disease are not likely induced by 6B antigens. Moreover, immunogenicity data from previous studies indicate that serotype 9V in PCVs does not cross-react with serotype 9N [56]. Furthermore, early immunological studies have shown that sera with serotype 23F slightly react with serotype 23A, but do not react with serotype 23B [57,58]. The structures of serotypes 23A, 23B, and 23F, which have been previously described, may explain these interactions, since serotype 23A and serotype 23F are similar in structure, while serotype 23B lacks an immunodominant α-rhamnose side chain that serotypes 23A and 23B both possess [57].

There are some limitations in this study. First, specimen collection or culturing practices could have changed over time and thereby influencing the number of pneumococcal positive isolates identified. Second, we were only able to ascertain trends for certain (but not all) serotypes within a serogroup. Third, the years of data available varied by country, so the time horizon of 1998 to 2019 was used in the analysis, as it was the most inclusive. Fourth, given that IPD is rare, and its cases can vary from year to year, annual cases of specific serotypes may be inconsistent. Therefore, the exact case counts might be less relevant compared to the overall trends observed for these serotypes. Fifth, this study did not assess the impact of vaccine administration schedule (e.g., 3 + 1, 2 + 1), which may likely influence cross-protection and immunity. Lastly, the number of countries reporting trends for VRTs was limited, so it was not possible to reach conclusions for some of the VRTs.

## 5. Conclusions

Our study indicates that while the use of different higher valency vaccines has contributed to the reduction in or sustained control of VT IPD, their impact against VRT cases differs, likely due to different serotypes included in each vaccine. Understanding the degree of cross-protection provided by different PCVs is crucial as it can affect changes in disease trends due to the type of pneumococcal vaccine used and should be taken into consideration when assessing pneumococcal vaccination programs. Given the role of VRT cases in the overall burden of IPD, countries should consider the potential impact of PCVs on VT and VRT disease when making pneumococcal vaccine decisions.

## Figures and Tables

**Figure 1 vaccines-13-00651-f001:**
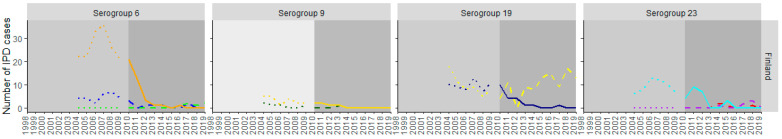
Trends in IPD cases attributable to serogroups 6, 9, 19, and 23 after PCV10 introduction. For Brazil, Chile, Finland, and Paraguay, only PCV10 (medium gray) was introduced during the study period. For Colombia, PCV7 was the first PCV implemented (medium gray), followed by PCV10 (dark gray). Brazil, Chile, Colombia, and Paraguay did not have data available for serogroups 9 and 23.

**Figure 2 vaccines-13-00651-f002:**
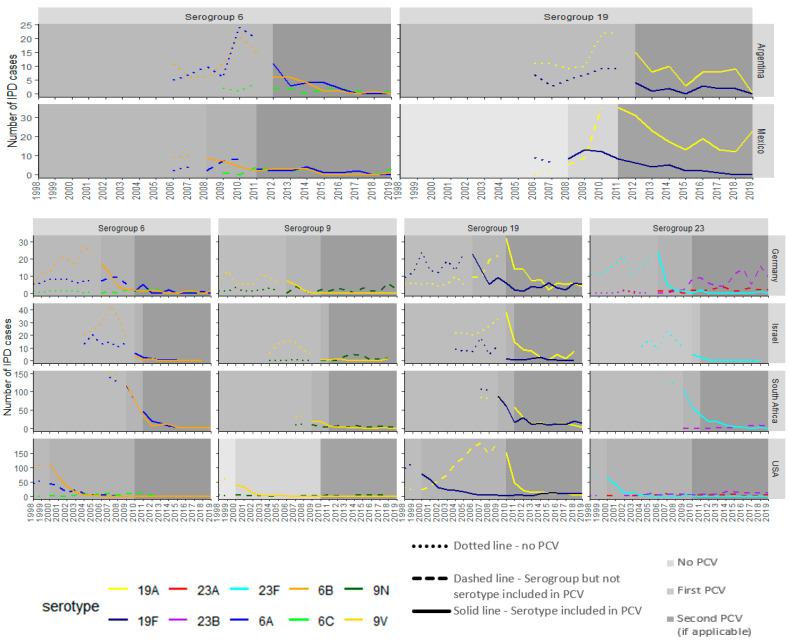
Trends in IPD cases attributable to serogroups 6, 9, 19, and 23 after PCV13 introduction. In Germany, both PCV10 and PCV13 are licensed and recommended; however, PCV13 is used primarily with >90% uptake in NIP. For Germany, Israel, Mexico, South Africa, and USA, PCV7 (medium gray) and PCV13 (dark gray) were implemented. In Argentina, only PCV13 (dark gray) was introduced. Argentina and Mexico did not have data for serogroups 9 and 23.

**Table 1 vaccines-13-00651-t001:** Countries included in analysis.

Country	First PCV and Posology Used in NIP	Year of First PCV Introduction	Second PCV and Posology Used in NIP	Second PCV Year of Introduction During Study Period	Years of Data Available	Serotype Data Available
Colombia	PCV7 (2 + 1)	2009	PCV10 (2 + 1)	2012	2006–2018	6A, 6B, 6C, 19A, 19F
Chile *	PCV10 (3 + 1)	2011	-	-	2006–2018	6A, 6B, 6C, 19A, 19F
Finland	PCV10 (2 + 1)	2010	-	-	2004–2019	6A, 6B, 6C, 9N, 9V, 19A, 19F, 23A, 23B, 23F
Brazil	PCV10 (3 + 1)	2010	-	-	2006–2018	6A, 6B, 6C, 19A, 19F
Paraguay	PCV10 (2 + 1)	2011	-	-	2006–2018	6A, 6B, 6C, 19A, 19F
Germany **	PCV7 (3 + 1)	2006	PCV13 (2 + 1)	2010	1998–2019	6A, 6B, 6C, 9N, 9V, 19A, 19F, 23A, 23B, 23F
Israel	PCV7 (2 + 1)	2009	PCV13 (2 + 1)	2010	2004–2019	6A, 6B, 6C, 9N, 9V, 19A, 19F
USA	PCV7 (3 + 1)	2000	PCV13 (3 + 1)	2010	1998–2019	6A, 6B, 6C, 9N, 9V, 19A, 19F, 23A, 23B, 23F
Mexico	PCV7 (3 + 0)	2008	PCV13 (2 + 1)	2011	2006–2018	6A, 6B, 6C, 19A, 19F
South Africa	PCV7 (2 + 1)	2009	PCV13 (2 + 1)	2011	2009–2019	6A, 6B, 6C, 9N, 9V, 19A, 19F, 23B, 23F
Argentina	PCV13 (2 + 1)	2012	-	-	2006–2018	6A, 6B, 6C, 19A, 19F

* Chile currently uses PCV13, which was introduced in 2017, but was using PCV10 during the study period. ** In Germany, both PCV10 and PCV13 are licensed and recommended; however, PCV13 is used primarily with >90% uptake in NIP. Data provided by Professor Mark Van Der Linden. NIP: National Immunization Program; PCV: Pneumococcal Conjugate Vaccine.

**Table 2 vaccines-13-00651-t002:** Classification of serotypes included in analysis as vaccine-type (VT) or vaccine-related type (VRT).

Serogroup	Serotypes	PCV7	PCV10	PCV13
6	6A	VRT	VRT	VT
	6B	VT	VT	VT
	6C	VRT	VRT	VRT
9	9N	VRT	VRT	VRT
	9V	VT	VT	VT
19	19A	VRT	VRT	VT
	19F	VT	VT	VT
23	23A	VRT	VRT	VRT
	23B	VRT	VRT	VRT
	23F	VT	VT	VT

## Data Availability

The raw data supporting the conclusions of this article will be made available by the authors on request.

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
