# Peer review of "Real-World Impact of Pneumococcal Conjugate Vaccines on Vaccine Serotypes and Potential Cross-Reacting Non-Vaccine Serotypes"

_vaccines, 2025, doi:10.3390/vaccines13060651_

Round 1

Reviewer 1 Report

Comments and Suggestions for Authors

Please include the following comments in the revised version of the manuscript

Specific comments:

  1. The paper has a good reason for studying vaccine-related serotypes (VRTs), but it should more clearly say that the goal is to explore ideas (not prove them) in both the Introduction and Discussion.
  2. The paper says later on that it can’t prove cause and effect, but this should be mentioned right away in the abstract or methods section.
  3. The paper should better explain how the related serotypes (VRTs and VTs) are connected  for example, whether they share similar structures to help readers understand why these serotypes were chosen.
  4. Since different countries used different vaccines at different times, it makes comparisons tricky. The authors could separate countries into groups or do extra checks to make the analysis more reliable.
  5. The study shows patterns in the data but doesn’t test them statistically. Even simple tests or models would help support the conclusions more strongly.
  6. The trend graphs are useful, but they don’t show exact numbers or variation over time. Adding these would make the graphs easier to understand.
  7. The paper has a clear conclusion, but it should say more about what this means in real life for example, how continued circulation of VRTs could affect vaccine decisions.

Reviewer 2 Report

Comments and Suggestions for Authors

I read the results of your study with great interest, as they have practical significance and provide insights into the existence of cross-immunity between vaccine and non-vaccine serotypes in the currently used pneumococcal conjugate vaccines (PCVs), PCV10 and PCV13, with reference to PCV7. 

A limitation of the study, as you have also noted, is that the analysis included only 11 countries that met the criterion of having used either PCV10 or PCV7/PCV13 exclusively. The challenge lies in the fact that many countries have adopted mixed vaccination models or have switched between vaccines in their immunization calendars, which makes real-life effects somewhat different from those described in the study.

Additionally, the vaccine administration schedule (e.g., 3+1, 2+1, or another model) was not specified, which could also influence the level of immunity and cross-protection. If this aspect was analyzed and valid data are available, it would be beneficial to include it, as the vaccine administration schedule is an important factor considered by countries when selecting vaccines for their national immunization programs.

In summary, I believe it would be useful to include this point in the commentary.

There are also some minor technical issues, such as spacing errors in several parts of the text, which should be corrected.

Reviewer 3 Report

Comments and Suggestions for Authors

The aim of the study was to examine whether available data suggest that different pneumococcal conjugate vaccines (PCVs) can reduce the incidence of invasive pneumococcal disease caused by cross-reactive serotypes. The researchers sought to assess this by examining trends in childhood invasive pneumococcal disease caused by cross-reactive vaccine-related serotypes in countries that have introduced PCV10 or PCV13 into their national vaccination programmes. Cross-reactivity could potentially extend vaccine protection to non-included serotypes. The team discusses the trend-like results obtained with caution. At the end of the manuscript, several limitations are mentioned, the list of which could be expanded at length (e.g., the antibiotic policy of the countries involved is unknown, which may affect the number of invasive pneumococcal diseases, and cross-protection between serotypes is often only assumed at the immunological level ((e.g., based on OPA values)), speculative, but the actual disease reduction does not always justify this). Nevertheless, I believe that understanding cross-reactivity is crucial for evaluating the full effectiveness of PCVs and for the development of future vaccines aimed at broader protection. Based on these, I support the publication of the article.

Major point:

The author mentions data from 13 countries in the abstract, but only 11 countries are mentioned in the manuscript. Please explain why. Maybe two of the 13 were excluded, or was there simply a typo?

Minor point:

Streptococcus pneumoniae should also be italicized among the keywords

Round 2

Reviewer 1 Report

Comments and Suggestions for Authors

Thank you for the corrections